# A linear algebra-based approach to understanding the relation between the winding number and zero-energy edge states

**Chen-Shen Lee**⋆

Department of Physics, National Taiwan University, Taipei 10617, Taiwan

⋆ 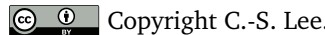 yausan0523270@gmail.com

## Abstract

The one-to-one relation between the winding number and the number of robust zero-energy edge states, known as bulk-boundary correspondence, is a celebrated feature of $1d$ systems with chiral symmetry. Although this property can be explained by the K-theory, the underlying mechanism remains elusive. Here, we demonstrate that, even without resorting to advanced mathematical techniques, one can prove this correspondence and clearly illustrate the mechanism using only Cauchy's integral and elementary algebra. Furthermore, our approach to proving bulk-boundary correspondence also provides clear insights into a kind of system that doesn't respect chiral symmetry but can have robust left or right zero-energy edge states. In such systems, one can still assign the winding number to characterize these zero-energy edge states.

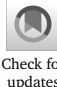
# 1 Introduction

Topological insulators are among the most exotic materials because they manifest the nontrivial boundary phenomena associated with topology [1–6]. One renowned class within topological insulators is that of strong topological insulators. Their non-trivial boundary phenomena are protected by on-site symmetries. More specifically, for a strong topological insulator with a given on-site symmetry in a non-trivial phase, an adiabatic deformation that preserves this on-site symmetry cannot break its non-trivial boundary phenomena without closing the bulk gap (i.e., without undergoing a phase transition). From this perspective, the strong topological insulators are in the family of symmetry-protected topological (SPT) phases [7,8]. The classification of SPT phases of strong topological insulators can be accomplished through the K-theory [9–11], with topological invariants that can detect these phases (see [12] and references therein). In this work, we focus on $1d$ free fermion systems with chiral symmetry, recognized as strong topological insulators classified by $\mathbb{Z}$, with the SPT phases detectable through the winding number. A well-known example is the Su-Schrieffer-Heeger (SSH) model, which describes a $1d$ chain of polyacetylene [13].

 The one-to-one relation between the bulk topological invariants and the non-trivial boundary phenomena is known as bulk-boundary correspondence [1–6,14–22]. Although this property has been numerically and experimentally confirmed in various studies, we don't have general proof of bulk-boundary correspondence. For our focus, $1d$ free fermion systems with chiral symmetry, bulk-boundary correspondence in these systems asserts that the number of robust zero-energy edge states is characterized by the winding number [12]. Despite the bulk-boundary correspondence here can be elucidated through certain advanced approaches, such as the K-theory [23] and connecting topological invariants to Green's functions [24], a transparent method to comprehend the underlying mechanism is still absent. This work aims to provide a legible and rudimentary proof of this correspondence. Specifically, we will show that the bulk-boundary correspondence can be proved by applying Cauchy's integral and elementary algebra. While several studies have utilized similar ideas to illustrate the relation between the winding number and zero-energy edge states [25–28], only bulk-boundary correspondence in two-band systems was rigorously proved through this method [28]. Here, by introducing the matrix pencils and matrix difference equations, we demonstrate that this approach can strictly and effectively prove bulk-boundary correspondence in systems with chiral symmetry, including multi-band cases. On the other hand, this transparent proof also specifies that if, by choosing a basis, the Bloch Hamiltonian of a given system can be brought into the

following forms

$$\mathscr{H}(k) = \begin{pmatrix} D_X(k) & D(k) \\ D^\dagger(k) & 0 \end{pmatrix}, \quad \text{or} \quad \mathscr{H}(k) = \begin{pmatrix} 0 & D(k) \\ D^\dagger(k) & D_Y(k) \end{pmatrix},$$

where $D_X(k)$ and $D_Y(k)$ can be any Hermitian matrix, the system has robust left or right zero-energy edge states characterized by the winding number $\nu(D^\dagger)$.

## 2  Chiral symmetry and the winding number

Conventionally, free fermion systems can be described in the single-particle basis set, yielding single-particle Hamiltonians $H$. Assuming translation symmetry, we can transform the single-particle Hamiltonian $H$ into the Bloch Hamiltonian $\mathscr{H}(k)$ through Fourier transformation. A system is said to respect chiral symmetry if there exists a unitary operator $S$, where

$$S\mathscr{H}(k)S^{-1} = -\mathscr{H}(k). \tag{1}$$

In the presence of chiral symmetry, the Hamiltonian can be brought into block-off diagonal form in the chiral basis, such as

$$\mathscr{H}(k) = \begin{pmatrix} 0 & D(k) \\ D^\dagger(k) & 0 \end{pmatrix}, \quad \text{with} \quad S = \begin{pmatrix} \mathbb{1} & 0 \\ 0 & -\mathbb{1} \end{pmatrix}, \tag{2}$$

where $D(k)$ has the dimension $n \times n$ and $\mathbb{1}$ is the $n \times n$ identity matrix. For $1d$ gapped systems with chiral symmetry, the winding number is defined as

$$
\begin{aligned}
\nu(D^\dagger) &= \frac{1}{2\pi i} \int_{BZ} dk \, \mathrm{Tr}[(D^\dagger)^{-1} \partial_k D^\dagger] \\
&= \frac{1}{2\pi i} \int_{BZ} dk \, \partial_k \log(\det[D^\dagger]),
\end{aligned} \tag{3}
$$

where $BZ$ denotes the first Brillouin zone, the notation Tr is trace, and det represents the determinant.

## 3  Systems with half nearest-neighbor hopping

Let's start with the simplest cases, the systems with half nearest-neighbor hopping, which means, in the chiral basis, the matrix $D^\dagger(k)$ of these systems has only the constant terms and $e^{ik}$ terms or the constant terms and $e^{-ik}$ terms. Here, we first focus on the previous one, that is,

$$D^\dagger(k) = A + Be^{ik}, \tag{4}$$

where $A$ and $B$ are $n \times n$ matrices. Note that $A$ and $B$ can be singular matrices, but $D(k)$ must be a non-singular matrix because the winding number (3) requires that $D(k)$ is invertible over Brillouin zone.

### 3.1  Winding number of systems with $D^\dagger(k) = A + Be^{ik}$

By rewriting $z = e^{ik}$, we have $D^\dagger(z) = A + Bz$, and $\det[D^\dagger]$ can be written as

$$\det[D^\dagger] = \det[A + Bz] = \sum_{n'=0}^{n'_r} a_{n'} z^{n'}, \tag{5}$$

where $n'_r \in \mathbb{Z}^{0+}$ and $a_{n'}$ is a complex coefficient. As shown in eq. (5), $\det[D^\dagger]$ can be read as a complex polynomial, so we can further factorize it as

$$\det[A + Bz] = f(z) = a_{n'_r} \prod_p (z - \eta_p)^{m_p}.\tag{6}$$

Here, $\eta_p$ are the roots of $\det[D^\dagger]$, and $m_p \in \mathbb{Z}^{0+}$ are the corresponding multiplicities. By replacing $\det[D^\dagger]$ in the winding number (3) with $\det[D^\dagger]$ shown in eq. (6), we have

$$\nu(A + Bz) = \frac{1}{2\pi i} \oint_{|z|=1} dz \, \frac{f'(z)}{f(z)} = \sum_p \frac{1}{2\pi i} \oint_{|z|=1} dz \, \frac{m_p}{z - \eta_p}.\tag{7}$$

In the above equation, we use the integration by substitution, $z = e^{ik}$ and $dz = ie^{ik}dk$. With the help of Cauchy's integral, the winding number can be simplified as

$$\nu(A + Bz) = \sum_{p, |\eta_p| < 1} m_p.\tag{8}$$

Therefore, the winding number $\nu$ can be interpreted as the count of multiplicities for all roots with an absolute value less than 1. Also, the above equation implies $\nu \geq 0$ here. It is worth noting that, as a prerequisite for the winding number (3), requiring $D^\dagger$ to be invertible over $k \in BZ$, ensures that $|\eta_p| \neq 1$ holds for all the roots in eq. (6).

## 3.2 Analytical calculation of robust zero-energy edge states of systems with $D^\dagger(k) = A + Be^{ik}$

To investigate the zero-energy edge states, we employ the inverse Fourier transformation and truncate the system without breaking the unit cells, which converts the Bloch Hamiltonian with $D^\dagger(k) = A + Be^{ik}$ into the following real space Hamiltonian

$$H = \sum_{j=0}^{N_c-1} [B |S_X, j+1\rangle \langle S_Y, j| + A |S_X, j\rangle \langle S_Y, j| + h.c.],\tag{9}$$

where $j$ is the cell index and $N_c$ is the number of unit cells. $|S_X\rangle$ and $|S_Y\rangle$ denote the basis states of the chiral operator $S$ shown in eq. (2) with eigenvalue $+1$ and with eigenvalue $-1$ respectively, and $|S_X/S_Y, j\rangle = |S_X/S_Y\rangle \otimes |j\rangle$. To be more intuitive, let us write $H$ in the matrix form

$$H = \begin{pmatrix} 0 & A & 0 & 0 & 0 & 0 & \cdots & \cdots & \cdots & 0 \\ A^\dagger & 0 & B^\dagger & 0 & 0 & 0 & \cdots & \cdots & \cdots & 0 \\ 0 & B & 0 & A & 0 & 0 & \cdots & \cdots & \cdots & 0 \\ 0 & 0 & A^\dagger & 0 & B^\dagger & 0 & \cdots & \cdots & \cdots & 0 \\ \vdots & & & & \ddots & & & & & \\ \vdots & & & & 0 & B & 0 & A & 0 & 0 \\ \vdots & & & & 0 & 0 & A^\dagger & 0 & B^\dagger & 0 \\ \vdots & & & & 0 & 0 & 0 & B & 0 & A \\ 0 & 0 & 0 & \cdots & 0 & 0 & 0 & 0 & A^\dagger & 0 \end{pmatrix}.\tag{10}$$

$H$ is a $(2n \cdot N_c) \times (2n \cdot N_c)$ matrix, where $n$ is the dimension of matrix $D^\dagger$. We should suppose the system is in the thermodynamic limit $N_c \to \infty$, implying a consideration of the semi-infinite

system. Otherwise, exact zero-energy edge states may not exist due to finite-size effects. Let's start with the right semi-infinite chain. By introducing

$$\psi = \sum_{a,j} c_X^{a,j} \left| S_X^a, j \right\rangle + c_Y^{a,j} \left| S_Y^a, j \right\rangle = (\left| X, 0 \right\rangle, \left| Y, 0 \right\rangle, ..., \left| X, N_c - 1 \right\rangle, \left| Y, N_c - 1 \right\rangle)^T,$$

where $c_X^{a,j}$ and $c_Y^{a,j}$ are complex coefficients, $\left| S_X/S_Y, j \right\rangle = \bigotimes_a \left| S_X^a/S_Y^a, j \right\rangle$, and $\left| X/Y, j \right\rangle = (c_{X/Y}^{1,j}, c_{X/Y}^{2,j}, ..., c_{X/Y}^{n,j})^T$ is the vector composed of the coefficients according to the basis $\left| S_X/S_Y, j \right\rangle$, the eigenvalue problem $H\psi = \epsilon\psi$ can be written as two series

$$\begin{aligned} B \left| Y, j-1 \right\rangle + A \left| Y, j \right\rangle &= \epsilon \left| X, j \right\rangle, \quad \text{with} \quad \left| Y, -1 \right\rangle = 0, \\ A^\dagger \left| X, j \right\rangle + B^\dagger \left| X, j+1 \right\rangle &= \epsilon \left| Y, j \right\rangle. \end{aligned} \tag{11}$$

Since we only focus on the zero-energy edge states, we can impose $\epsilon = 0$ on the above equation, which leads to

$$\begin{aligned} B \left| Y, j \right\rangle + A \left| Y, j+1 \right\rangle &= 0, \quad \text{with} \quad \left| Y, -1 \right\rangle = 0, \\ A^\dagger \left| X, j \right\rangle + B^\dagger \left| X, j+1 \right\rangle &= 0. \end{aligned} \tag{12}$$

We will have a discussion on equations including boundary conditions in the next section, so let's temporarily ignore the first equation in eq. (12). To proceed, let us introduce an operator $\Delta$ defined as $\Delta \left| X, j \right\rangle = \left| X, j+1 \right\rangle$, and then we have

$$(A^\dagger + \Delta B^\dagger) \left| X, j \right\rangle = 0. \tag{13}$$

Without loss of generality, to find the edge states, we can employ the standard ansatz, $\Delta = \zeta$ and $\left| X, j \right\rangle = \left| X \right\rangle \zeta^j$ where $\zeta$ is a complex number, which turns eq. (13) into

$$(A^\dagger + \zeta B^\dagger) \left| X \right\rangle = 0. \tag{14}$$

The above equation is also called the generalized eigenvalue problem. For the non-trivial solutions where $\left| X \right\rangle$ is not a zero vector, we have

$$\det[A^\dagger + \zeta B^\dagger] = \det[A^\dagger + \eta^* B^\dagger] = (\det[A + \eta B])^* = [f(\eta)]^* = 0. \tag{15}$$

Here we assume $\zeta = \eta^*$, where the symbol $*$ denotes the complex conjugate. Also, for a matrix $M$, there is a relation $\det[M^\dagger] = (\det[M])^*$. Now, the connection between the winding number and the zero-energy edge states becomes apparent. All the roots $\eta_p$ in eq. (6) can contribute the solutions $\zeta = \eta_p^*$ to eq. (15), implying that the zero-energy edge states are given by

$$\left| X, j \right\rangle = \left| X \right\rangle (\eta_p^*)^j, \quad |\eta_p^*| < 1. \tag{16}$$

The condition $|\eta_p^*| < 1$ is necessary (i.e., only the left zero-energy edge states are valid), otherwise $\left| X, j \right\rangle$ will diverge when considering the thermodynamic limit. The above discussion already indicated bulk-boundary correspondence in the systems that possess only non-zero $\eta_p$ with multiplicity 1.[1] Now, let's dive into the cases with $\eta_p = 0$ and degenerated roots, starting with the definition of matrix pencils (see [29] for more information)

$$(A, B) = A - \lambda B, \quad \text{with} \quad A, B \in \mathbb{C}^{m \times n}, \tag{17}$$

where $\lambda$ is indeterminate. A matrix pencil $(A, B)$ is said to be regular if $n = m$ and there is $\lambda \in \mathbb{C}$ such that $(A, B)$ is invertible. In this sense, since we require $D^\dagger(k)$ over $k \in BZ$ to be

---

[1]The same argument was made in [25]. They used a similar approach to prove bulk-boundary correspondence in systems with a more relaxed assumption. They supposed all roots are non-degenerate based on a pragmatic route. Also, they didn't discuss the cases with zero roots.

invertible, we can regard $(A^\dagger + \zeta B^\dagger) = (A^\dagger - (-\zeta)B^\dagger) = (A^\dagger, B^\dagger)$ as a regular matrix pencil. For a regular matrix pencil, the eigenvalues are defined as

$$\begin{array}{l} 1. \text{ The roots of } p(\lambda) = \det[A - \lambda B],\\ 2. \; \infty \text{ with multiplicity } n - \deg[p(\lambda)], \end{array} \tag{18}$$

where $A$ and $B$ are $n \times n$ matrix, and $\deg[p(\lambda)]$ denotes the degree of polynomial $p(\lambda)$. Weierstrass (1867) laid a foundation for studying regular matrix pencils. He proved that, for a given regular matrix pencil, there are two non-singular matrices, $P$ and $Q$, such that

$$P(A - \lambda B)Q = \text{diag}\{L_{\lambda_1}, \ldots, L_{\lambda_a}, N_{\lambda_\infty}\}, \tag{19}$$

where

$$\begin{aligned} L_{\lambda_i} &= -\lambda \mathbb{1}_{n_i \times n_i} + J_{n_i}(\lambda_i),\\ N_{\lambda_\infty} &= -\lambda J_{n_\infty}(0) + \mathbb{1}_{n_\infty \times n_\infty}. \end{aligned} \tag{20}$$

Here, $J_{n_i}(\lambda_i)$ is a $n_i \times n_i$ Jordan block with eigenvalue $\lambda_i$, and $n_\infty$ is the multiplicity of $\lambda_i = \infty$. In some literature (e.g., [30]), the diagonal form in eq. (19) is called Weierstrass canonical form. Note that, except for $\lambda_i = \infty$, $n_i$ is not determined by the multiplicity of $\lambda_i$. Depending on the situation, an eigenvalue with multiplicity $m_i$ may contribute $1 \sim m_i$ Jordan blocks to eq. (19), where the direct sum of these Jordan blocks forms a $m_i \times m_i$ matrix. We now go back to our topic. Because $(A^\dagger, B^\dagger)$ is a regular matrix pencil, where the finite eigenvalues and their algebraic multiplicities are given by the negative of the complex conjugate of the roots in eq. (6) and their multiplicities, we can turn eq. (14) into

$$P(A^\dagger + \eta^* B^\dagger)QQ^{-1}|X\rangle = \text{diag}\{L_{-\eta_1^*}, \ldots, L_{-\eta_a^*}, N_{-\eta_\infty^*}\}|X'\rangle = 0, \tag{21}$$

where $\zeta = \eta^*$ and $|X'\rangle = Q^{-1}|X\rangle$. The technique used in the above equation is fairly common when addressing differential-algebraic systems of equations (see [31] for more information). Obviously, eq. (21) can be separated into two parts. One part dominated by $N_{\eta_\infty^*}$ just tells us some elements in $|X'\rangle$ are zero. The other part regarding $L_{\eta_i}$ forms a matrix difference equation

$$\text{diag}\{L_{-\eta_1^*}, \ldots, L_{-\eta_a^*}\}|X_L'\rangle = 0 \rightarrow |X_L', j+1\rangle = M_J|X_L', j\rangle, \tag{22}$$

where $M_J = \text{diag}\{J_{n_1}(-\eta_1^*) \ldots, J_{n_a}(-\eta_a^*)\}$ is a $\deg[p(\eta^*)] \times \deg[p(\eta^*)]$ matrix and $|X_L'\rangle$ is defined by $|X'\rangle = \{|X_L'\rangle, |X_N'\rangle\}^T$. Here, $|X_N'\rangle$ is a $n_\infty \times 1$ zero vector, which is determined by $N_{\eta_\infty^*}$. In the above equation, we turn the ansatz back to $|X'\rangle(\eta^*)^j = |X', j\rangle$. Now, the answer to why $\eta_p^*$ with multiplicity $m_p$ represents $m_p$ linearly independent edge states is clear. Let's consider the systems without $\eta_p^* = 0$ first. Recall that, for the difference equation $\mathbf{u}_{j+1} = M\mathbf{u}_j$, where $M$ is a $n \times n$ matrix, there are $n$ linearly independent solutions, such as

$$\mathbf{u}_j = M^j \mathbf{u}_0 = P_M J_M^j P_M^{-1} \mathbf{u}_0, \tag{23}$$

where $J_M$ is the Jordan normal form of $M$, and $P_M$ denotes the corresponding generalized modal matrix that consists of the eigenvectors and generalized eigenvectors. As a case in point, consider a $3 \times 3$ matrix $M$ with two distinct eigenvalues $\lambda_1$ and $\lambda_2$, where $\lambda_1$ has an algebraic multiplicity of 2 but a geometric multiplicity of 1. Assuming the corresponding eigenvectors are $\mathbf{v}$ and $\mathbf{w}$ for $\lambda_1$ and $\lambda_2$, and the generalized eigenvector is defined as $(M - \lambda_1 \mathbb{1})\mathbf{v}' = \mathbf{v}$, we

have

$$\mathbf{u}_j = M^j \mathbf{u}_0$$

$$= \{\mathbf{v}, \mathbf{v}', \mathbf{w}\} \begin{pmatrix} \lambda_1 & 1 & 0 \\ 0 & \lambda_1 & 0 \\ 0 & 0 & \lambda_2 \end{pmatrix}^j \{\mathbf{v}, \mathbf{v}', \mathbf{w}\}^{-1} \mathbf{u}_0$$

$$= \{\mathbf{v}, \mathbf{v}', \mathbf{w}\} \begin{pmatrix} \lambda_1^j & j\lambda_1^{j-1} & 0 \\ 0 & \lambda_1^j & 0 \\ 0 & 0 & \lambda_2^j \end{pmatrix} \begin{pmatrix} c_1 \\ c_2 \\ c_3 \end{pmatrix} \tag{24}$$

$$= c_1 \lambda_1^j \mathbf{v} + c_2 (j\lambda_1^{j-1} \mathbf{v} + \lambda_1^j \mathbf{v}') + c_3 \lambda_2^j \mathbf{w},$$

where $\{\mathbf{v}, \mathbf{v}', \mathbf{w}\}^{-1} \mathbf{u}_0 = \{c_1, c_2, c_3\}^T$. It's clear that there are three linearly independent solutions. Therefore, for the matrix difference equation (22), we have the following comment:

> If $\eta_p^* \neq 0$ is an eigenvalue of $M_J$ and its algebraic multiplicity is $m_p^*$, we have $m_p^*$ linearly independent solutions that behave as $|X_L'\rangle (\eta^*)^j$, where $|X_L'\rangle$ are composed of the corresponding eigenvectors and generalized eigenvectors. $\qquad$ (25)

Now, let's go into the details of the zero roots. The zero root with multiplicity $m_0$ in eq. (6) means that $M_J$ has zero eigenvalue with algebraic multiplicity $m_0$. If the geometric multiplicity of $\eta_i^* = 0$ is also $m_0$, we have $m_0$ linearly independent eigenvectors where

$$M_J |X_L', j\rangle = |X_L', j+1\rangle = 0, \quad \text{for } j \geq 0. \tag{26}$$

The above equation implies that $|X_L', j\rangle$ are suddenly truncated for $j \geq 1$, but we still have non-zero $|X_L', 0\rangle$ that comprises of $m_0$ linearly independent eigenvectors, such as

$$|X_L', 0\rangle = \sum_{i=1}^{m_0} c_i \mathbf{u}_{0,i}, \tag{27}$$

where $\mathbf{u}_{0,i}$ represent the eigenvectors with $\eta_i^* = 0$. It will be more tricky if the geometric multiplicity of $\eta_i^* = 0$ is not $m_0$. For simplicity, we start the discussion with an example where $m_0 = 2$ and the corresponding geometric multiplicity is 1. Suppose the eigenvector with $\eta_i^* = 0$ of this example is $\mathbf{u}_{0,1}$, then the generalized eigenvector $\mathbf{u}_{0,2}$ is given by

$$M_J \mathbf{u}_{0,2} = \mathbf{u}_{0,1}, \tag{28}$$

where $\mathbf{u}_{0,1}$ and $\mathbf{u}_{0,2}$ are linearly independent. Given that $\mathbf{u}_{0,1}$ is the eigenvector with $\eta_i^* = 0$, we have $(M_J)^2 \mathbf{u}_{0,2} = M_J \mathbf{u}_{0,1} = 0$. Also, combined with the fact that $M_J |X_L', j\rangle = |X_L', j+1\rangle$, the above equation actually means that, when $|X_L', 0\rangle = \mathbf{u}_{0,2}$, we have $|X_L', 1\rangle = \mathbf{u}_{0,1}$ and $|X_L', j\rangle = 0$ for $j \geq 2$. Therefore, this case contains two linearly independent solutions,

$$c_1 (|X_L', 0\rangle = \mathbf{u}_{0,1}) + c_2 (|X_L', 0\rangle = \mathbf{u}_{0,2} + |X_L', 1\rangle = \mathbf{u}_{0,1}). \tag{29}$$

That is to say, there is a solution that has both $|X_L', 0\rangle$ and $|X_L', 1\rangle$ non-zero. To better illustrate this, we provide an example in Appendix A. By extending the idea used in this example, we can make the following statement:

> If $\eta_p^* = 0$ with algebraic multiplicity $m_p^*$ is the eigenvalue of $M_J$, it contributes $m_p^*$ linearly independent solutions to eq. (22). These solutions, consisting of the corresponding eigenvectors and generalized eigenvectors, are suddenly truncated somewhere, with the upper limit of truncated cell indices set at $j = m_p^* - 1$. $\qquad$ (30)

Therefore, all the eigenvectors and generalized eigenvectors of $M_J$ serve as the composition of the solutions to eq. (22).

Taking statements (25) and (30) into account, the number of non-trivial linearly independent solutions of the matrix difference equation (22) is $\sum_p m_p^*$, where $m_p^*$ is the multiplicity of the root $\eta_p^*$. The number of non-trivial linearly independent solutions of eq. (13) is also $\sum_p m_p^*$ because the invertible transformation $\left|X'\right\rangle = Q^{-1}\left|X\right\rangle$ doesn't break linear independence.[2] Additionally, as stated before, we should exclude the solutions with $\eta_p^* > 1$ because they are divergent as $j \to \infty$. Therefore, combining with the fact that each converged non-trivial linearly independent solution of eq. (13) represents a robust left zero-energy edge state situated on $\left|S_X, j\right\rangle$, we can conclude that

> For right semi-infinite chains, the number of robust left zero-energy edge states located on $\left|S_X, j\right\rangle$ is $\sum_{p, |\eta_p^*| < 1} m_p^*$. (31)

Note that we have $m_p^* = m_p$, which comes from eq. (15), and $|\eta_p^*| = |\eta_p|$. Finally, comparing with the winding number (8), we can see the bulk-boundary correspondence is established.

Although the above discussion is for right semi-infinite chains, we can adopt the same idea for left semi-infinite chains. First, we relabel the cell indices from the right-hand side, leading to the eigenvectors of $H$ expressed as $(\left|X, N_c - 1\right\rangle, \left|Y, N_c - 1\right\rangle, \ldots, \left|X, 0\right\rangle, \left|Y, 0\right\rangle)^T$. Then, the eigenvalue problem of $H$ can be written as two series

$$B^\dagger \left|X, j-1\right\rangle + A^\dagger \left|X, j\right\rangle = \epsilon \left|Y, j\right\rangle, \quad \text{with} \quad \left|X, -1\right\rangle = 0,$$
$$A \left|Y, j\right\rangle + B \left|Y, j+1\right\rangle = \epsilon \left|X, j\right\rangle. \tag{32}$$

The above equations can be readily obtained by observing eq. (10). By adopting the ansatz $\left|Y, j\right\rangle = \left|Y\right\rangle \zeta^j$, the corresponding robust zero-energy edge states can be given by solving

$$(A + \zeta B)\left|Y\right\rangle = 0, \tag{33}$$

where $\zeta$ is a complex number. By going through the same process as discussed right semi-infinite chains, we can connect the non-trivial solutions of eq. (33) to the roots in eq. (6) as $\zeta = \eta_p$ and then make the following statement:

> For left semi-infinite chains, the number of robust right zero-energy edge states located on $\left|S_Y, j\right\rangle$ is $\sum_{p, |\eta_p| < 1} m_p$. (34)

In short, for the cases with $D^\dagger(k) = A + Be^{ik}$, the winding number $\nu$ is identical to the number of robust left (right) zero-energy edge states situated on $\left|S_X, j\right\rangle$ ($\left|S_Y, j\right\rangle$) when considering right (left) semi-infinite chains.

## 3.3 Discussion on systems with $D^\dagger(k) = A + Ce^{-ik}$

Now, let's move to the other type of systems with half nearest-neighbor hopping, where the matrix $D^\dagger(k)$ is read as

$$D^\dagger(k) = A + Ce^{-ik}. \tag{35}$$

---

[2]With the invertible transformation $\left|X'\right\rangle = Q^{-1}\left|X\right\rangle$, the zero-energy edge states located on $\left|S_X, j\right\rangle$ can be determined by $(\left|X, 0\right\rangle, \left|Y, 0\right\rangle, \ldots, \left|X, N_c - 1\right\rangle, \left|Y, N_c - 1\right\rangle)^T = (Q\left|X', 0\right\rangle, \left|Y, 0\right\rangle, \ldots, Q\left|X', N_c - 1\right\rangle, \left|Y, N_c - 1\right\rangle)^T$, where all $\left|Y, j\right\rangle$ are zero vectors because the zero-energy edge states located on $\left|S_X, j\right\rangle$ and $\left|S_Y, j\right\rangle$ are separately determined by two equations as shown in eq. (12).

Unlike the substitution we used before, here we utilize $z = e^{-ik}$, which leads to

$$\det[D^\dagger] = \det[A + Cz] = \sum_{n=0}^{n_r} a_n z^n . \tag{36}$$

By factorizing the above complex polynomial, we have

$$\det[A + Cz] = f(z) = a_{n_r} \prod_p (z - \eta_p)^{m_p} . \tag{37}$$

With the substitution $z = e^{-ik}$ and $dz = -ie^{-ik}dk$, the corresponding winding number is given by

$$\nu(A + Cz) = \frac{1}{2\pi i} \oint_{|z|=1} dz \, \frac{f'(z)}{f(z)} = \sum_j \frac{1}{2\pi i} \oint_{|z|=1} dz \, \frac{m_p}{z - \eta_p} . \tag{38}$$

Simplifying the above equation by Cauchy's integral, we can get

$$\nu(A + Cz) = -\sum_{p,|\eta_p|<1} m_p . \tag{39}$$

The above equation implies $\nu \leq 0$.

For studying the zero-energy edge states, we convert the Bloch Hamiltonian into the following real space Hamiltonian

$$H = \sum_{j=0}^{N_c-1} [A |S_X, j\rangle \langle S_Y, j| + C |S_X, j\rangle \langle S_Y, j+1| + h.c.], \tag{40}$$

or

$$H = \begin{pmatrix} 0 & A & 0 & C & 0 & 0 & \cdots & \cdots & \cdots & 0 \\ A^\dagger & 0 & 0 & 0 & 0 & 0 & \cdots & \cdots & \cdots & 0 \\ 0 & 0 & 0 & A & 0 & C & \cdots & \cdots & \cdots & 0 \\ C^\dagger & 0 & A^\dagger & 0 & 0 & 0 & \cdots & \cdots & \cdots & 0 \\ \vdots & & & & \ddots & & & & & \\ \vdots & & & & 0 & 0 & 0 & A & 0 & C \\ \vdots & & & & C^\dagger & 0 & A^\dagger & 0 & 0 & 0 \\ \vdots & & & & 0 & 0 & 0 & 0 & 0 & A \\ 0 & 0 & 0 & \cdots & 0 & 0 & C^\dagger & 0 & A^\dagger & 0 \end{pmatrix} . \tag{41}$$

If we consider the right semi-infinite chains, the eigenvalue problem is read as

$$\begin{aligned} A |Y, j\rangle + C |Y, j+1\rangle &= \epsilon |X, j\rangle , \\ C^\dagger |X, j-1\rangle + A^\dagger |X, j\rangle &= \epsilon |Y, j\rangle , \quad \text{with} \quad |X, -1\rangle = 0 . \end{aligned} \tag{42}$$

For the left semi-infinite chains, we have

$$\begin{aligned} A^\dagger |X, j\rangle + C^\dagger |X, j+1\rangle &= \epsilon |Y, j\rangle , \\ C |Y, j-1\rangle + A |Y, j\rangle &= \epsilon |X, j\rangle , \quad \text{with} \quad |Y, -1\rangle = 0 . \end{aligned} \tag{43}$$

Let's still ignore the equations with boundary conditions. In the next section, we'll elaborate on why the equations with boundary conditions here don't contribute any robust zero-energy edge states. As stated before, the robust zero-energy edge states can be given by solving

$$\begin{aligned} (A + \zeta C) |Y\rangle &= 0, && \text{for right semi-infinite chains,} \\ (A^\dagger + \zeta C^\dagger) |X\rangle &= 0, && \text{for left semi-infinite chains.} \end{aligned} \tag{44}$$



The bulk-boundary correspondence here can be investigated by the same method when discussing $D^\dagger(k) = A + Be^{ik}$, which leads to the following statement: for the cases with $D^\dagger(k) = A + Ce^{-ik}$, the absolute value of winding number $|\nu|$ is identical to the number of robust left (right) zero-energy edge states situated on $|S_Y, j\rangle$ ($|S_X, j\rangle$) when considering right (left) semi-infinite chains.

## 4 Systems with nearest-neighbor hopping

In the chiral basis, the general form of the matrix $D^\dagger(k)$ for systems with nearest-neighbor hopping can be written as

$$D^\dagger(k) = Ce^{-ik} + A + Be^{ik}, \tag{45}$$

where $A$, $B$, and $C$ are $n \times n$ matrices and can be singular matrices, but $D(k)$ must be invertible.

### 4.1 Winding number of systems with nearest-neighbor hopping

We have two choices of substitution, $z_+ = e^{ik}$ and $z_- = e^{-ik}$. With $z_+ = e^{ik}$, $\det[D^\dagger]$ can be written as

$$\det[D^\dagger] = \det[Cz_+^{-1} + A + Bz_+] = z_+^{-n}\det[C + Az_+ + Bz_+^2] = z_+^{-n}\sum_{n'=0}^{n'_{+,r}} a_{n'}z_+^{n'}, \tag{46}$$

where $n'_{+,r} \in \mathbb{Z}^{0+}$ and $a_{n'}$ is a complex coefficient. The above equation can be further factorized as

$$\det[Cz_+^{-1} + A + Bz_+] = z_+^{-n}f_1(z_+), \tag{47}$$

where

$$f_1(z_+) = \det[C + Az_+ + Bz_+^2] = a_{n'_{+,r}}\prod_p (z_+ - \eta_{+,p})^{m_p^+}. \tag{48}$$

Here, $\eta_{+,p}$ are the roots of $f_1(z_+)$, and $m_p^+$ are the corresponding multiplicities. By using this substitution and $dz_+ = ie^{ik}dk$, the winding number can be read as

$$\nu(Cz_+^{-1} + A + Bz_+) = -n + \sum_p \frac{1}{2\pi i}\oint_{|z_+|=1} dz_+ \frac{m_p^+}{z_+ - \eta_{+,p}}. \tag{49}$$

Utilizing Cauchy's integral can simplify the winding number as

$$\nu(Cz_+^{-1} + A + Bz_+) = -n + \sum_{p,|\eta_{+,p}|<1} m_p^+. \tag{50}$$

For the other choice $z_- = e^{-ik}$, we have

$$\det[D^\dagger] = \det[Cz_- + A + Bz_-^{-1}] = z_-^{-n}\det[B + Az_- + Cz_-^2] = z_-^{-n}\sum_{n'=0}^{n'_{-,r}} b_{n'}z_-^{n'}, \tag{51}$$

where $n'_{-,r} \in \mathbb{Z}^{0+}$ and $b_{n'}$ is a complex coefficient. After factorizing, the above equation becomes

$$\det[Cz_- + A + Bz_-^{-1}] = z_-^{-n}g_1(z_-), \tag{52}$$

where

$$g_1(z_-) = \det[B + A z_- + C z_-^2] = b_{n'_{-,r}} \prod_p (z_- - \eta_{-,p})^{m_p^-} . \tag{53}$$

$\eta_{-,p}$ are the roots of $g_1(z_-)$, and $m_p^-$ are the corresponding multiplicities. With $dz_- = -i e^{ik} dk$ and the help of Cauchy's integral, we have

$$\nu(C z_- + A + B z_-^{-1}) = n - \sum_{p,|\eta_{-,p}|<1} m_p^- . \tag{54}$$

Note that we just employ different substitutions to rewrite $D^\dagger(k) = C e^{-ik} + A + B e^{ik}$, so the winding numbers in eq. (48) and eq. (53) must be the same, which leads to

$$\sum_{p,|\eta_{+,p}|<1} m_p^+ + \sum_{p,|\eta_{-,p}|<1} m_p^- = 2n . \tag{55}$$

The above equation implies a connection between $f_1(z_+)$ and $g_1(z_-)$, which will play a crucial role in the following discussion.

## 4.2 Analytical calculation of robust zero-energy edge states for systems with nearest-neighbor hopping

By truncating systems without breaking unit cells after inverse Fourier transformation, the systems with zero-energy edge states can be described as

$$H = \sum_{j=0}^{N_c-1} [B\,|S_X, j+1\rangle\,\langle S_Y, j| + A\,|S_X, j\rangle\,\langle S_Y, j| + C\,|S_X, j\rangle\,\langle S_Y, j+1| + h.c.], \tag{56}$$

or

$$H = \begin{pmatrix} 0 & A & 0 & C & 0 & 0 & \cdots & \cdots & \cdots & 0 \\ A^\dagger & 0 & B^\dagger & 0 & 0 & 0 & \cdots & \cdots & \cdots & 0 \\ 0 & B & 0 & A & 0 & C & \cdots & \cdots & \cdots & 0 \\ C^\dagger & 0 & A^\dagger & 0 & B^\dagger & 0 & \cdots & \cdots & \cdots & 0 \\ \vdots & & & & \ddots & & & & & \\ \vdots & & & & & 0 & B & 0 & A & 0 & C \\ \vdots & & & & & C^\dagger & 0 & A^\dagger & 0 & B^\dagger & 0 \\ \vdots & & & & & 0 & 0 & 0 & B & 0 & A \\ 0 & 0 & 0 & \cdots & 0 & 0 & C^\dagger & 0 & A^\dagger & 0 \end{pmatrix} . \tag{57}$$

For right semi-infinite chains, the eigenvalue problem of the above systems can be read as

$$\begin{aligned} B\,|Y, j-1\rangle + A\,|Y, j\rangle + C\,|Y, j+1\rangle &= \epsilon\,|X, j\rangle , \quad \text{with} \quad |Y, -1\rangle = 0 , \\ C^\dagger\,|X, j-1\rangle + A^\dagger\,|X, j\rangle + B^\dagger\,|X, j+1\rangle &= \epsilon\,|Y, j\rangle , \quad \text{with} \quad |X, -1\rangle = 0 . \end{aligned} \tag{58}$$

The corresponding zero-energy edge states can be given by solving

$$\begin{aligned} B\,|Y, j-1\rangle + A\,|Y, j\rangle + C\,|Y, j+1\rangle &= 0 , \quad \text{with} \quad |Y, -1\rangle = 0 , \\ C^\dagger\,|X, j-1\rangle + A^\dagger\,|X, j\rangle + B^\dagger\,|X, j+1\rangle &= 0 , \quad \text{with} \quad |X, -1\rangle = 0 , \end{aligned} \tag{59}$$

or, equivalently

$$\begin{aligned} \begin{pmatrix} A & B \\ \mathbb{1} & 0 \end{pmatrix} L_{Y,j} &= \begin{pmatrix} -C & 0 \\ 0 & \mathbb{1} \end{pmatrix} L_{Y,j+1} , \qquad \text{with} \quad |Y, -1\rangle = 0 , \\ \begin{pmatrix} A^\dagger & C^\dagger \\ \mathbb{1} & 0 \end{pmatrix} L_{X,j} &= \begin{pmatrix} -B^\dagger & 0 \\ 0 & \mathbb{1} \end{pmatrix} L_{X,j+1} , \qquad \text{with} \quad |X, -1\rangle = 0 , \end{aligned} \tag{60}$$

where $L_{X,j} = \{|X,j\rangle, |X,j-1\rangle\}^T$, $L_{Y,j} = \{|Y,j\rangle, |Y,j-1\rangle\}^T$, and $\mathbb{1}$ is the $n \times n$ identity matrix. With the assumptions, $L_{X,j} = L_X \zeta^j$ and $L_{Y,j} = L_Y \zeta^j$, we can turn the above equations into matrix pencils, where the finite eigenvalues are determined by

$$\det\left[\begin{pmatrix} A & B \\ \mathbb{1} & 0 \end{pmatrix} - \begin{pmatrix} -C\zeta & 0 \\ 0 & \mathbb{1}\zeta \end{pmatrix}\right] = (-1)^n \det[B + A\zeta + C\zeta^2] = (-1)^n g_1(\zeta) = 0\,,$$

$$\det\left[\begin{pmatrix} A^\dagger & C^\dagger \\ \mathbb{1} & 0 \end{pmatrix} - \begin{pmatrix} -B^\dagger\zeta & 0 \\ 0 & \mathbb{1}\zeta \end{pmatrix}\right] = (-1)^n \det[C^\dagger + A^\dagger\zeta + B^\dagger\zeta^2] = (-1)^n [f_1(\zeta^*)]^* = 0\,. \tag{61}$$

If we don't consider the boundary conditions, by using the same fashion as in the previous section, we'll find that every unique eigenvalue $\zeta_j$ of the matrix pencils in eq. (61) contributes $m_j$, which is the algebraic multiplicity of $\zeta_j$, non-trivial linearly independent solutions to the corresponding equations in eq. (60). Moreover, the eigenvalues of the first matrix pencil in eq. (61) are identical to $\eta_{-,p}$, and for the other matrix pencil, the eigenvalues are given by $\eta_{+,p}^*$. Therefore, with the restriction $|\zeta_j| < 1$, we have

Without the boundary conditions:

the number of the linearly independent solutions $L_{Y,j}$ is $\displaystyle\sum_{p,|\eta_{-,p}|<1} m_p^-\,,$

the number of the linearly independent solutions $L_{X,j}$ is $\displaystyle\sum_{p,|\eta_{+,p}|<1} m_p^+\,.$ $\tag{62}$

Now, let's discuss the role of boundary conditions. After inserting these linearly independent solutions into eq. (60), the boundary condition can be written as

$$L_{Y,0} = \begin{pmatrix} |Y,0\rangle \\ 0 \end{pmatrix} = \sum_{n'} c_{n'} \mathbf{v}_{n'}(j=0)\,,$$

$$L_{X,0} = \begin{pmatrix} |X,0\rangle \\ 0 \end{pmatrix} = \sum_{m'} c_{m'} \mathbf{v}_{m'}(j=0)\,, \tag{63}$$

where $\{\mathbf{v}_{n'}(j)\}$ and $\{\mathbf{v}_{m'}(j)\}$ represent the set of linearly independent non-trivial solutions with cell indices $j$ for $L_{Y,j}$ and $L_{X,j}$, respectively. Obviously, given that the 0 parts in the above equations are $n \times 1$ zero vectors, they will eliminate $n$ linear independence of these solutions **at most**. Also, considering the fact that every non-trivial linearly independent solution in eq. (60) means a zero-energy edge state located on the corresponding basis, here we draw the following conclusion first and will explain more.

> 1. If $n < \sum_{p,|\eta_{-,p}|<1} m_p^-$, there are $|\nu| = -n + \sum_{p,|\eta_{-,p}|<1} m_p^-$ **robust** left zero-energy edge states located on $|S_Y, j\rangle$.
>
> 2. If $n < \sum_{p,|\eta_{+,p}|<1} m_p^+$, there are $\nu = -n + \sum_{p,|\eta_{+,p}|<1} m_p^+$ **robust** left zero-energy edge states located on $|S_X, j\rangle$. $\tag{64}$

To be more specific, because the boundary conditions don't always provide $n$ restrictions (of the coefficients), it is possible that a system has zero-energy edge states not characterized by the winding number. However, these zero-energy edge states are not robust, and we call them **trivial** zero-energy edge states here. We provide an example of a system with **trivial** zero-energy edge states in Appendix. B. **Trivial** zero-energy edge states can be destroyed by turning on other hopping terms, which can be regarded as perturbations, without breaking chiral symmetry and closing the bulk gap (i.e., without changing topology). But note that,

no matter how we turn on hopping terms, the **maximum** number of restrictions given by boundary conditions here is always $n$. Therefore, the number of **robust** zero-energy edge states is determined by supposing that there are $n$ restrictions. Finally, by considering the relation between the roots of $g_1(z_-)$ and $f_1(z_+)$ as expressed in eq. (55), we can make the following statement

> For right semi-infinite chains, if $v \geq 0$, there are $v$ robust left zero-energy edge states located on $|S_X, j\rangle$, and if $v \leq 0$, there are $|v|$ robust left zero-energy edge states located on $|S_Y, j\rangle$. (65)

By using the same method as discussed above, for left semi-infinite chains, we have

> For left semi-infinite chains, if $v \geq 0$, there are $v$ robust right zero-energy edge states located on $|S_Y, j\rangle$, and if $v \leq 0$, there are $|v|$ robust right zero-energy edge states located on $|S_X, j\rangle$. (66)

## 5 Systems with arbitrary long-range hopping

Although we can choose a non-primitive unit cell that makes all hopping nearest-neighbor, which doesn't change the value of the winding number, the corresponding chiral symmetry is different from that of the primitive unit cell. Hence, choosing a non-primitive unit cell will lead to different SPT phases from selecting the primitive one. Taking this into account, discussion on systems with long-range hopping is necessary. Let's start with the following $D^\dagger(k)$ matrix

$$D^\dagger(k) = A + \sum_{n'=1}^{n_C} C_{n'} e^{-in'k} + \sum_{m'=1}^{n_B} B_{m'} e^{im'k}, \tag{67}$$

where $n_C, n_B \in \mathbb{Z}^{0+}$. The above $n \times n$ matrix $D^\dagger(k)$ can represent systems with arbitrary long-range hopping in the chiral basis. As the same requirement before, $A$, $B_{m'}$, and $C_{n'}$ can be singular, but $D^\dagger(k)$ must be invertible.

### 5.1 Winding number of systems with arbitrary long-range hopping

For the substitution $z_+ = e^{ik}$, we have

$$\det[D^\dagger] = z_+^{-n \cdot n_C} \det\left[ A z_+^{n_C} + \sum_{n'=1}^{n_C} C_{n'} z_+^{n_C - n'} + \sum_{m'=1}^{n_B} B_{m'} z_+^{m'+n_C} \right] = z_+^{-n \cdot n_C} f_2(z_+), \tag{68}$$

where the factorized part $f_2(z_+)$ can be written as

$$f_2(z_+) = \det\left[ A z_+^{n_C} + \sum_{n'=1}^{n_C} C_{n'} z_+^{n_C - n'} + \sum_{m'=1}^{n_B} B_{m'} z_+^{m'+n_C} \right] = c_+ \prod_p (z_+ - \eta_{+,p})^{m_p^+}. \tag{69}$$

$c_+$ is a complex number. With this substitution and Cauchy's integral, the winding number is given by

$$v(D^\dagger) = -(n \cdot n_C) + \sum_{p, |\eta_{+,p}| < 1} m_p^+. \tag{70}$$

We also can use the substitution $z_- = e^{-ik}$, which leads to

$$\det[D^\dagger] = z_-^{-n \cdot n_B} \det\left[ A z_-^{n_B} + \sum_{m'=1}^{n_B} B_{m'} z_-^{n_B - m'} + \sum_{n'=1}^{n_C} C_{n'} z_-^{n'+n_B} \right] = z_-^{-n \cdot n_B} g_2(z_-), \tag{71}$$

where

$$g_2(z_-) = \det\left[Az_-^{n_B} + \sum_{m'=1}^{n_B} B_{m'} z_-^{n_B-m'} + \sum_{n'=1}^{n_C} C_{n'} z_-^{n'+n_B}\right] = c_- \prod_p (z_- - \eta_{-,p})^{m_p^-}. \tag{72}$$

$c_-$ is a complex number. The corresponding winding number is

$$\nu(D^\dagger) = (n \cdot n_B) - \sum_{p,|\eta_{-,p}|<1} m_p^-. \tag{73}$$

Since different substitutions do not change the value of the winding number, there is a relation between the roots of $f_2(z_+)$ and $g_2(z_-)$

$$\sum_{p,|\eta_{+,p}|<1} m_p^+ + \sum_{p,|\eta_{-,p}|<1} m_p^- = n \cdot (n_B + n_C). \tag{74}$$

## 5.2 Analytical calculation of robust zero-energy edge states for systems with arbitrary long-range hopping

As the method used in the previous discussion, after inverse Fourier transformation, we can equip systems with zero-energy edge states by truncating them without breaking unit cells, such as

$$H = \sum_{j=0}^{N_c-1} \left[A|S_X,j\rangle\langle S_Y,j| + \sum_{m'=1}^{n_B} B_{m'}\left|S_X,j+m'\right\rangle\langle S_Y,j| + \sum_{n'=1}^{n_C} C_{n'}|S_X,j\rangle\left\langle S_Y,j+n'\right| + h.c.\right]. \tag{75}$$

For right semi-infinite chains, the corresponding eigenvalue problem can be written as

$$A|Y,j\rangle + \sum_{m'=1}^{n_B} B_{m'}\left|Y,j-m'\right\rangle + \sum_{n'=1}^{n_C} C_{n'}\left|Y,j+n'\right\rangle = \epsilon|X,j\rangle,$$

$$\text{with} \quad |Y,-1\rangle = |Y,-2\rangle\ldots|Y,-n_B\rangle = 0, \tag{76}$$

and

$$A^\dagger|X,j\rangle + \sum_{n'=1}^{n_C} C_{n'}^\dagger\left|X,j-n'\right\rangle + \sum_{m'=1}^{n_B} B_{m'}^\dagger\left|X,j+m'\right\rangle = \epsilon|Y,j\rangle,$$

$$\text{with} \quad |X,-1\rangle = |X,-2\rangle\ldots|X,-n_C\rangle = 0. \tag{77}$$

The zero-energy edge states can be given by solving the above two equations with $\epsilon = 0$, which can be equivalently written as the following generalized eigenvalue problems

$$\begin{aligned} M_Y L_{Y,j} &= D_Y L_{Y,j+1}, \quad &&\text{with} \quad |Y,-1\rangle = |Y,-2\rangle\ldots|Y,-n_B\rangle = 0, \\ M_X L_{X,j} &= D_X L_{Y,j+1}, \quad &&\text{with} \quad |X,-1\rangle = |X,-2\rangle\ldots|X,-n_C\rangle = 0, \end{aligned} \tag{78}$$

where

$$M_Y = \begin{pmatrix} C_{n_C-1} & C_{n_C-2} & \cdots & C_1 & A & B_1 & \cdots & B_{n_B-1} & B_{n_B} \\ \mathbb{1} & 0 & \cdots & \cdots & \cdots & \cdots & \cdots & \cdots & 0 \\ 0 & \mathbb{1} & 0 & \cdots & \cdots & \cdots & \cdots & \cdots & 0 \\ \vdots & & \ddots & & & & & & \vdots \\ \vdots & & & \ddots & & & & & \vdots \\ \vdots & & & & \ddots & & & & \vdots \\ \vdots & & & & & \ddots & & & \vdots \\ \vdots & & & & & & \ddots & & \vdots \\ 0 & \cdots & \cdots & \cdots & \cdots & \cdots & 0 & \mathbb{1} & 0 \end{pmatrix},$$

$$M_X = \begin{pmatrix} B_{n_B-1}^\dagger & B_{n_B-2}^\dagger & \cdots & B_1^\dagger & A^\dagger & C_1^\dagger & \cdots & C_{n_C-1}^\dagger & C_{n_C}^\dagger \\ \mathbb{1} & 0 & \cdots & \cdots & \cdots & \cdots & \cdots & \cdots & 0 \\ 0 & \mathbb{1} & 0 & \cdots & \cdots & \cdots & \cdots & \cdots & 0 \\ \vdots & & \ddots & & & & & & \vdots \\ \vdots & & & \ddots & & & & & \vdots \\ \vdots & & & & \ddots & & & & \vdots \\ \vdots & & & & & \ddots & & & \vdots \\ \vdots & & & & & & \ddots & & \vdots \\ 0 & \cdots & \cdots & \cdots & \cdots & \cdots & 0 & \mathbb{1} & 0 \end{pmatrix},$$

(79)

$$D_Y = \begin{pmatrix} -C_{n_C} & 0 & \cdots & \cdots & 0 \\ 0 & \mathbb{1} & 0 & \cdots & 0 \\ \vdots & & \ddots & & \vdots \\ \vdots & & & \ddots & 0 \\ 0 & \cdots & \cdots & 0 & \mathbb{1} \end{pmatrix}, \quad D_X = \begin{pmatrix} -B_{n_B}^\dagger & 0 & \cdots & \cdots & 0 \\ 0 & \mathbb{1} & 0 & \cdots & 0 \\ \vdots & & \ddots & & \vdots \\ \vdots & & & \ddots & 0 \\ 0 & \cdots & \cdots & 0 & \mathbb{1} \end{pmatrix}, \quad (80)$$

and

$$L_{Y,j} = \{|Y, j+n_C-1\rangle, \cdots, |Y, j+1\rangle, |Y, j\rangle, |Y, j-1\rangle \cdots |Y, j-n_B\rangle\}^T,$$
$$L_{X,j} = \{|X, j+n_B-1\rangle, \cdots, |X, j+1\rangle, |X, j\rangle, |X, j-1\rangle \cdots |X, j-n_C\rangle\}^T.$$

(81)

Here, $\mathbb{1}$ is the $n \times n$ identity matrix, so $M_X$, $M_Y$, $D_X$, and $D_Y$ are $[n \cdot (n_C + n_B)] \times [n \cdot (n_C + n_B)]$ matrices. With the ansatz $L_{X,j} = L_X \zeta^j$ and $L_{Y,j} = L_Y \zeta^j$, the equations in eq. (81) become matrix pencils, and the corresponding finite eigenvalues are determined by

$$\det(M_Y - \zeta D_Y) = (-1)^{[n \cdot (n_C+n_B-1)]} g_2(\zeta) = 0,$$
$$\det(M_X - \zeta D_X) = (-1)^{[n \cdot (n_C+n_B-1)]} [f_2(\zeta^*)]^* = 0.$$

(82)

The derivation of eq. (82) is provided in Appendix. C. Therefore, we have

Without the boundary conditions:

The number of the linearly independent solutions $L_{Y,j}$ is $\displaystyle\sum_{p,|\eta_{-,p}|<1} m_p^-$,

the number of the linearly independent solutions $L_{X,j}$ is $\displaystyle\sum_{p,|\eta_{+,p}|<1} m_p^+$.

(83)

By imposing the boundary conditions in eq. (78) on $L_{Y,0}$ ($L_{X,0}$), which gives $n \cdot n_B$ ($n \cdot n_C$) equations of boundary conditions at most, we can make the following statement:

> 1. If $n \cdot n_B < \sum_{p,|\eta_{-,p}|<1} m_p^-$, there are $|\nu| = -n \cdot n_B + \sum_{p,|\eta_{-,p}|<1} m_p^-$ **robust** left zero-energy edge states located on $|S_Y, j\rangle$.
>
> 2. If $n \cdot n_C < \sum_{p,|\eta_{+,p}|<1} m_p^+$, there are $\nu = -n \cdot n_C + \sum_{p,|\eta_{+,p}|<1} m_p^+$ **robust** left zero-energy edge states located on $|S_X, j\rangle$.

(84)

Note that, although the boundary conditions also impose certain restrictions on $L_{Y,j>0}$ ($L_{X,j>0}$), by definition, the restricted components of $L_{Y,j>0}$ ($L_{X,j>0}$) must be the components of $L_{Y,0}$ ($L_{X,0}$), so there is no new equation of boundary conditions given by $L_{Y,j>0}$ ($L_{X,j>0}$). Lastly, combined with eq. (74), we can see bulk-boundary correspondence established as eq. (65). Also, utilizing the above approach to discussing left semi-infinite chains will lead to the bulk-boundary correspondence described in eq. (66).

## 6 Robust zero-energy edge states in the systems without chiral symmetry

In the previous sections, we proved that all robust zero-energy edge states have non-zero amplitude only on one of $|X, j\rangle$ and $|Y, j\rangle$. This property supports a sort of system that doesn't respect chiral symmetry but can have robust zero-energy edge states characterized by the winding number $\nu(D^\dagger)$, which is described by

$$\mathscr{H}(k) = \begin{pmatrix} D_X(k) & D(k) \\ D^\dagger(k) & 0 \end{pmatrix}, \quad \text{or} \quad \mathscr{H}(k) = \begin{pmatrix} 0 & D(k) \\ D^\dagger(k) & D_Y(k) \end{pmatrix}. \tag{85}$$

The only requirement here is $D_X(k) = D_X^\dagger(k)$ and $D_Y(k) = D_Y^\dagger(k)$, which guarantees the system is Hermitian. Let's focus on systems with $D_X(k)$ first, which can be decomposed into

$$\mathscr{H}(k) = \mathscr{H}_C(k) + \mathscr{H}_X(k), \quad \text{with} \quad \mathscr{H}_C(k) = \begin{pmatrix} 0 & D(k) \\ D^\dagger(k) & 0 \end{pmatrix}, \quad \text{and} \quad \mathscr{H}_X(k) = \begin{pmatrix} D_X(k) & 0 \\ 0 & 0 \end{pmatrix}. \tag{86}$$

The corresponding real space Hamiltonian can be written as

$$H = H_C + H_X, \tag{87}$$

where $H_C$ comes from the inverse Fourier transformation of $\mathscr{H}_C(k)$ with suitable truncation (i.e., truncation without breaking unit cells) and can be described as $H$ in eq. (75). $H_X$ corresponds to $\mathscr{H}_X(k)$ and can be read as

$$H_X = \sum_{j=0}^{N_c-1} \left[ \sum_{n'=0}^{n_T} T_{n'} |S_X, j+n'\rangle \langle S_X, j| + h.c. \right]. \tag{88}$$

Because $H_C$ respects chiral symmetry, it possesses robust zero-energy edge states characterized by the winding number. Depending on the right or left semi-infinite limit we consider and the value of the winding number, the robust zero-energy edge states of $H_C$ can be $\psi_X^e$ or $\psi_Y^e$, where $\psi_X^e$ and $\psi_Y^e$ represent the robust zero-energy edge states located on $|S_X, j\rangle$ and $|S_Y, j\rangle$, respectively. For all the zero-energy edge states $\psi_Y^e$, we have

$$H\psi_Y^e = H_C\psi_Y^e + H_X\psi_Y^e = 0. \tag{89}$$

Therefore, all the robust zero-energy edge states of $H_C$ located on $|S_Y, j\rangle$ are also robust zero-energy edge states of $H$. The above equation is established because of $H_C \psi_Y^e = 0$ and $H_X \psi_Y^e = 0$. The fact that $\psi_Y^e$ are "zero-energy" edge states of $H_C$ leads to $H_C \psi_Y^e = 0$. The other relation $H_X \psi_Y^e = 0$ is given by $\langle S_X, j | S_Y, j' \rangle = 0$. Lastly, combined with eq. (65) and (66), we can establish bulk-boundary correspondence in the systems with non-zero $D_X(k)$, such as

> For the Bloch Hamiltonian of a given system that can be written as $\mathscr{H}(k) = \mathscr{H}_C(k) + \mathscr{H}_X(k)$ in a certain basis, we can still assign the winding number $\nu$ defined in eq. (3) to this system. If $\nu \geq 0$, it has $\nu$ robust right zero-energy edge states located on $|S_Y, j\rangle$ when considering left semi-infinite chains. If $\nu \leq 0$, it has $|\nu|$ robust left zero-energy edge states located on $|S_Y, j\rangle$ when considering right semi-infinite chains. (90)

This correspondence can be intuitively grasped. Given that these robust zero-energy edge states live only on the $S_Y$ sublattice, any alteration limited to the $S_X$ sublattice sites, even if it breaks chiral symmetry, will not affect them.

Also, we can employ the same idea to study

$$\mathscr{H}(k) = \mathscr{H}_C(k) + \mathscr{H}_Y(k), \quad \text{with} \quad \mathscr{H}_C(k) = \begin{pmatrix} 0 & D(k) \\ D^\dagger(k) & 0 \end{pmatrix}, \quad \text{and} \quad \mathscr{H}_Y(k) = \begin{pmatrix} 0 & 0 \\ 0 & D_Y(k) \end{pmatrix},$$
(91)

which leads to the following bulk-boundary correspondence

> For the Bloch Hamiltonian of a given system that can be written as $\mathscr{H}(k) = \mathscr{H}_C(k) + \mathscr{H}_Y(k)$ in a certain basis, we can still assign the winding number $\nu$ defined in eq. (3) to this system. If $\nu \geq 0$, it has $\nu$ robust left zero-energy edge states located on $|S_X, j\rangle$ when considering right semi-infinite chains. If $\nu \leq 0$, it has $|\nu|$ robust right zero-energy edge states located on $|S_X, j\rangle$ when considering left semi-infinite chains. (92)

## 6.1 Example: A two-band model

As a concrete example, we consider a two-band system where its Bloch Hamiltonian is given by

$$\mathscr{H}_{2b}(k) = \begin{pmatrix} t_0 + t_1(e^{ik} + e^{-ik}) & u + we^{-ik} \\ u + we^{ik} & 0 \end{pmatrix}.$$
(93)

Since we want to show that $D_X(k)$ doesn't affect the existence of robust zero-energy edge states, we deliberately introduce the nearest-neighbor hopping $t_1$. The winding number of $\mathscr{H}_{2b}(k)$ can be read as

$$\nu(u + we^{ik}) = \begin{cases} 1, & w > u, \\ 0, & w < u. \end{cases}$$
(94)

After utilizing inverse Fourier transformation and then truncating the real space Hamiltonian without breaking unit cells, we obtain

$$H_{2b} = \sum_{j=0}^{N_c-1} [u |S_X, j\rangle \langle S_Y, j| + w |S_X, j+1\rangle \langle S_Y, j| + t_0 |S_X, j\rangle \langle S_X, j| + t_1 |S_X, j+1\rangle \langle S_X, j| + h.c.].$$
(95)

In accordance with the statement (90), when $w > u$, the corresponding left semi-infinite chain has a robust right zero-energy edge state located on $|S_Y, j\rangle$. The existence of this zero-energy

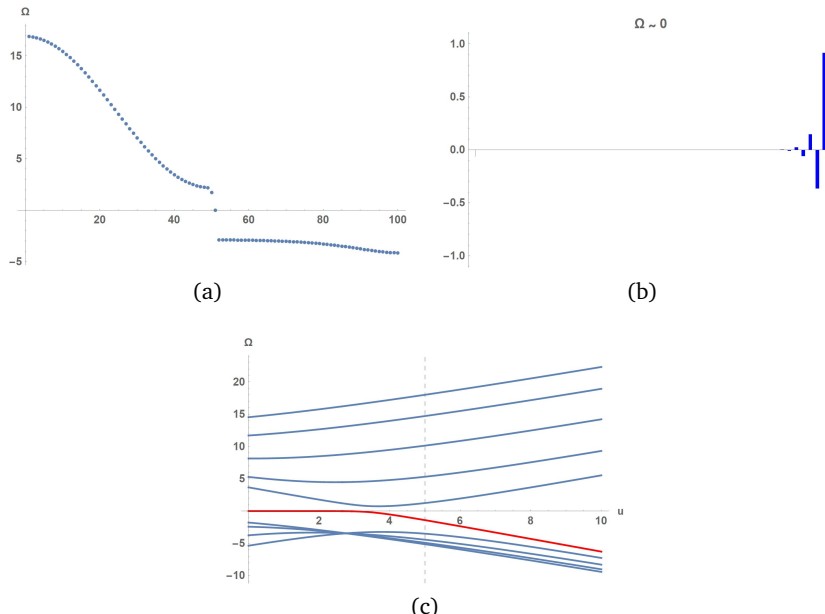

Figure 1: (a) The energy spectrum of $H_{2b}$ with $u = 2$, $w = 5$, $t_0 = 6$, and $t_1 = 4$. The number of unit cells is $N_c = 50$. (b) The zero-energy edge state of $H_{2b}$. The green and blue bars denote the value of wave functions on $|S_X, j\rangle$ and $|S_Y, j\rangle$, respectively. (c) The energy spectrum of $H_{2b}(u)$ with $w = 5$, $t_0 = 6$, and $t_1 = 4$. The number of unit cells is $N_c = 5$. The dashed line represents the point $u = w$, and the red color targets the deformation of the zero-energy edge state.

edge state can be numerically confirmed as shown in Fig. 1. Note that, due to the finite size effect, a finite but large enough system usually possesses only edge states with energy slightly splitting from zero, but the number and behavior of these edge states can be determined by combining the predictions from its corresponding right and left semi-infinite chains. Additionally, we further impose small spatial disorders on $H_{2b}$, such as

$$\tilde{H}_{2b} = H_{2b} + \delta H_{2b}, \tag{96}$$

where

$$\delta H_{2b} = \sum_{j=0}^{N_c-1} [\delta u(j) |S_X, j\rangle \langle S_Y, j| + \delta w(j) |S_X, j+1\rangle \langle S_Y, j| + \delta t_0(j) |S_X, j\rangle \langle S_X, j|$$
$$+ \delta t_1(j) |S_X, j+1\rangle \langle S_X, j| + h.c.]. \tag{97}$$

The numerical result in Fig. 2 indicates that the zero-energy edge state of $H_{2b}$ is robust against spatial disorders.

## 7 Conclusion

In this work, we commenced with the fact that for any Bloch Hamiltonian that respects chiral symmetry, its winding number can be linked to a complex polynomial $P_c$. More specifically, each root of $P_c$ with an absolute value less than 1 contributes its multiplicity to the winding number. Subsequently, by truncating the real-space Hamiltonian—derived through the inverse Fourier transformation of the Bloch Hamiltonian described in the chiral basis—without breaking unit cells, we equip the system with zero-energy edge states in the thermodynamic limit.

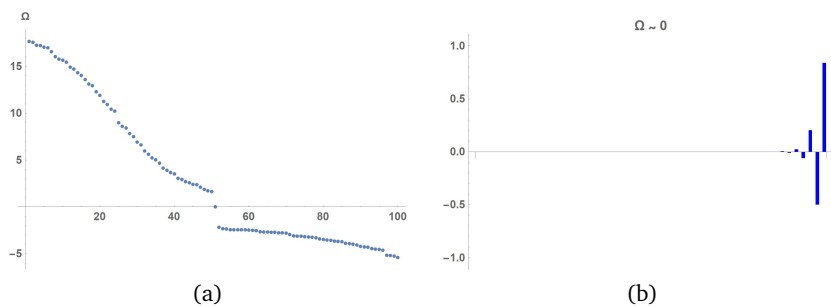

Figure 2: Here we introduce the spatial perturbations $\delta H_{2b}$. The perturbations $\delta u(j)$, $\delta w(j)$, $\delta t_0(j)$, and $\delta t_1(j)$ in each cell $j$ are within the range $[-1, 1]$. (a) The energy spectrum of $\tilde{H}_{2b}$ with $u = 2$, $w = 5$, $t_0 = 6$, and $t_1 = 4$. The number of unit cells is $N_c = 50$. (b) The zero-energy edge state of $\tilde{H}_{2b}$. The green and blue bars denote the value of wave functions on $|S_X, j\rangle$ and $|S_Y, j\rangle$, respectively.

These zero-energy edge states can be given by solving a regular matrix pencil, where the finite eigenvalues of this regular matrix pencil and their algebraic multiplicities are determined by the roots of $P_c$ and their multiplicities. After transforming this matrix pencil into Weierstrass canonical form, a matrix difference equation emerges. Since this transformation is an invertible linear transformation, each converged non-trivial linear independent solution of the corresponding matrix difference equation can represent a zero-energy edge state. Finally, by considering that the boundary conditions generate the largest number of linearly independent equations, we can see the number of remaining converged linearly independent solutions are characterized by the winding number as stated in eq. (65) and (66). Therefore, the bulk-boundary correspondence is established. It's worth noting that, because boundary conditions don't always impose the most restrictions, $1d$ systems with chiral symmetry may harbor zero-energy edge states not characterized by the winding number. We refer to these zero-energy edge states as trivial zero-energy edge states since they can be eliminated by introducing certain hopping terms without breaking chiral symmetry and closing the bulk gap.

On the other hand, owing to the property that all robust zero-energy edge states of semi-infinite systems with chiral symmetry are only situated on one of $|S_X, j\rangle$ and $|S_Y, j\rangle$, if the Bloch Hamiltonian of a given system that doesn't respect chiral symmetry can be written as the forms (85) in a certain basis, we can assign the winding number $\nu(D^\dagger)$ to characterize the robust zero-energy edge states of this system, which leads to the bulk-boundary correspondence described in the statements (90) and (92).

# Acknowledgments

I would like to thank Chang-Tse Hsieh and Hsien-Chung Kao for valuable discussions.

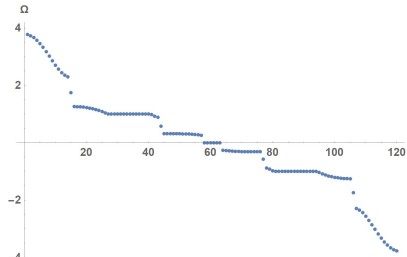

Figure 3: Energy spectrum of the real space Hamiltonian related to $D^\dagger$ in eq. (A.1), which is given by inverse Fourier transformation of the corresponding Bloch Hamiltonian and then truncating it without breaking unit cells. The number of unit cells is $N_c = 15$.

## A  Example of a system with zero roots

To show the argument regarding zero roots in eq. (30) holds everywhere, instead of some simple and obvious systems, we deliberately choose a complex system

$$D^\dagger(e^{ik}) = \begin{pmatrix} 1/2 & 1 & 1/2 & 1 \\ 0 & 0 & 0 & 0 \\ 1/2 & 0 & 1/2 & 0 \\ 0 & 0 & 0 & 1 \end{pmatrix} + \begin{pmatrix} 2 & 0 & 1 & 0 \\ 0 & 1 & 0 & 0 \\ 1 & 0 & 1 & 0 \\ 0 & 0 & 0 & 0 \end{pmatrix} e^{ik}. \tag{A.1}$$

This system cannot be easily constructed in reality. In fact, we establish it by $P^{-1}W(e^{ik})Q^{-1}$ $= D^\dagger(e^{ik})$ with specific $W$, which is the Weierstrass canonical form, and two casually chosen invertible matrices, $P$ and $Q$. After equipping it with zero-energy edge states and going through the process as discussed in Sec. 3, we obtain the corresponding matrix pencil $D^\dagger(\zeta)$. With two invertible matrices $P$ and $Q$, we can bring this matrix pencil into the Weierstrass canonical form $PD^\dagger(\zeta)Q = W(\zeta)$, where

$$P = \begin{pmatrix} 1 & 0 & -1 & -1 \\ 0 & 1 & 0 & 0 \\ 0 & 0 & 1 & 0 \\ 0 & 0 & 0 & 1 \end{pmatrix}, \quad Q = \begin{pmatrix} 1 & 0 & 0 & 0 \\ 0 & 1 & 0 & 0 \\ -1 & 0 & 1 & 0 \\ 0 & 0 & 0 & 1 \end{pmatrix}, \tag{A.2}$$

and

$$W = \begin{pmatrix} \zeta & 0 & 0 & 0 \\ 0 & \zeta & 0 & 0 \\ 0 & 0 & \zeta & 0 \\ 0 & 0 & 0 & 0 \end{pmatrix} + \begin{pmatrix} 0 & 1 & 0 & 0 \\ 0 & 0 & 0 & 0 \\ 0 & 0 & 1/2 & 0 \\ 0 & 0 & 0 & 1 \end{pmatrix}. \tag{A.3}$$

The above equation implies the eigenvalues of the matrix pencil $D(\zeta)$ are 0, 0, 1/2, and $\infty$. The non-trivial solutions of $W|X'\rangle = 0$ are determined by

$$\begin{pmatrix} \zeta & 0 & 0 \\ 0 & \zeta & 0 \\ 0 & 0 & \zeta \end{pmatrix} |X'_L\rangle = - \begin{pmatrix} 0 & 1 & 0 \\ 0 & 0 & 0 \\ 0 & 0 & 1/2 \end{pmatrix} |X'_L\rangle, \tag{A.4}$$

with $|X'\rangle = \{|X'_L\rangle, 0\}^T$. After turning the ansatz back to $|X'\rangle(\zeta)^j = |X', j\rangle$, we have

$$|X'_L, j+1\rangle = M_J |X'_L, j\rangle, \quad \text{where} \quad M_J = - \begin{pmatrix} 0 & 1 & 0 \\ 0 & 0 & 0 \\ 0 & 0 & 1/2 \end{pmatrix}. \tag{A.5}$$

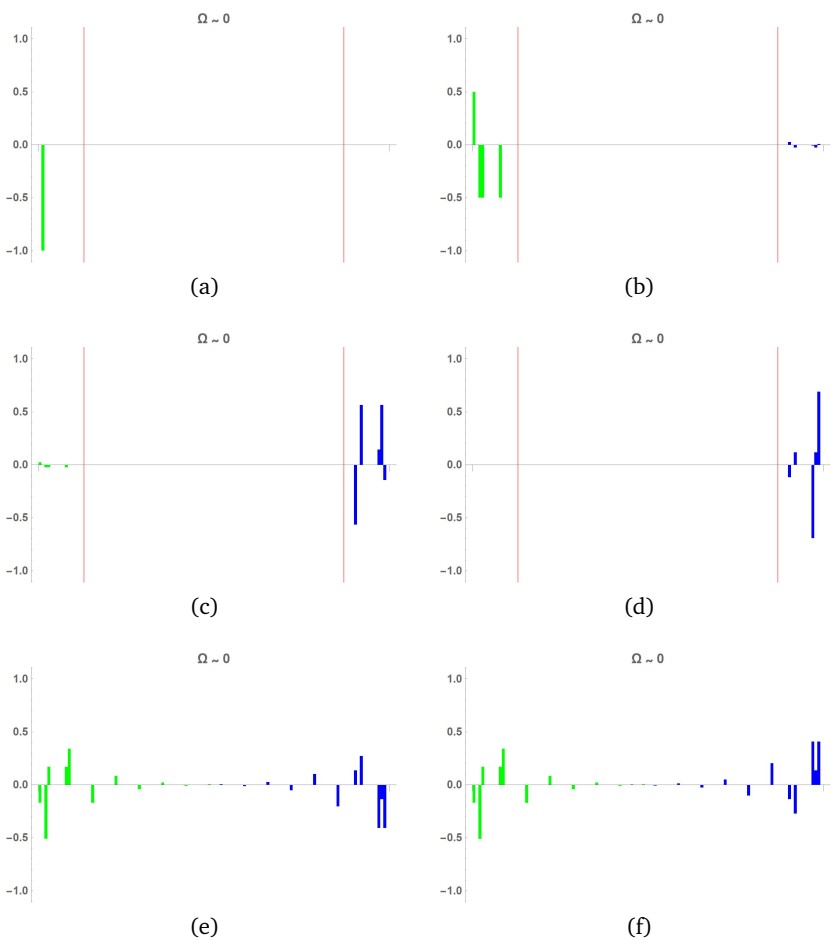

Figure 4: The (nearly) zero-energy edge states of the system in Fig. 3. The red lines represent the truncated cell index ($j = 1$), which is given by eq. (A.7), and the green and blue bars denote the value of wave functions on $|S_X, j\rangle$ and $|S_Y, j\rangle$, respectively. As expected, there are four hybridized suddenly truncated edge states and two hybridized exponentially decreasing edge states. Note that, for more complex systems, numerical results may give us edge states hybridizing suddenly truncated and exponentially decreasing edge states. However, we can "purify" them by linear superposition.

As stated in Sec. 3, the solutions of the above matrix difference equation are composed of the eigenvectors and generalized eigenvectors of $M_J$, so there are three linearly independent solutions, where one is exponentially decreasing, such as

$$\left| X'_L, j \right\rangle = c_1 \begin{pmatrix} 0 \\ 0 \\ 1 \end{pmatrix} (1/2)^j . \tag{A.6}$$

The other two are suddenly truncated, which is given by

$$\left| X'_L, j \right\rangle = c_2 \left[ \left| X'_L, 0 \right\rangle = \begin{pmatrix} 1 \\ 0 \\ 0 \end{pmatrix} \right] + c_3 \left[ \left| X'_L, 0 \right\rangle = \begin{pmatrix} 0 \\ 1 \\ 0 \end{pmatrix} + \left| X'_L, 1 \right\rangle = \begin{pmatrix} 1 \\ 0 \\ 0 \end{pmatrix} \right] . \tag{A.7}$$

Therefore, we can assert there are three left (right) zero-energy edge states located on $|S_X, j\rangle$ ($|S_Y, j\rangle$) for the corresponding right (left) semi-infinite chain, where two of them are suddenly

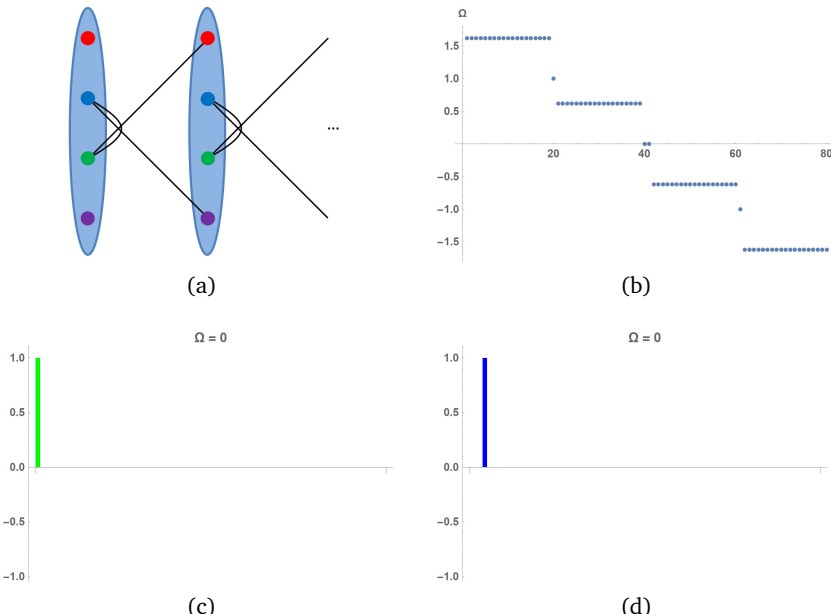

Figure 5: (a) The schematic picture of $H_{\text{tri}}(t = 0)$. (b) The energy spectrum of $H_{\text{tri}}(t = 0)$. The number of unit cells is $N_c = 20$. (c)(d) The trivial zero-energy edge states of $H_{\text{tri}}(t = 0)$. The green and blue bars denote the value of wave functions on $|S_X, j\rangle$ and $|S_Y, j\rangle$, respectively.

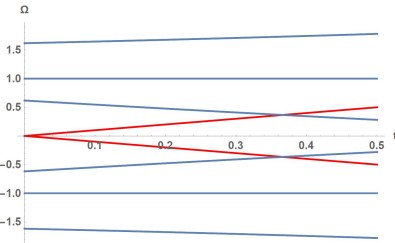

Figure 6: The energy spectrum of $H_{\text{tri}}(t)$ with $N_c = 20$. The zero-energy edge states of $H_{\text{tri}}(t = 0)$ vanish after turning on $t$.

truncated. The numerical results in Fig. 3 and Fig. 4 also agree with the above argument. It should be remarked that, for finite systems, we usually cannot see the exact zero-energy edge states, and the numerical result gives only edge states with energy slightly deviating from zero. Moreover, finite systems hybridize the predictions from the right and left semi-infinite chains, so the finite system related to eq. (A.1) possesses six edge states with energy around zero.

## B  System with trivial zero-energy edge states

Let's consider a system described by

$$D_{\text{tri}}^{\dagger}(k, t) = \begin{pmatrix} e^{ik} & t \\ 1 & e^{-ik} \end{pmatrix}. \tag{B.1}$$

The corresponding real-space Hamiltonian is given by

$$
H_{\text{tri}}(t) = \sum_{j=0}^{N_c-1} \left[ \begin{pmatrix} 0 & t \\ 1 & 0 \end{pmatrix} |S_X, j\rangle \langle S_Y, j| + \begin{pmatrix} 1 & 0 \\ 0 & 0 \end{pmatrix} |S_X, j+1\rangle \langle S_Y, j| \right.
$$
$$
\left. + \begin{pmatrix} 0 & 0 \\ 0 & 1 \end{pmatrix} |S_X, j\rangle \langle S_Y, j+1| + h.c. \right].
$$

(B.2)

Besides $t = 1$, where the system becomes gapless, the winding number is always zero. If we investigate $H_{\text{tri}}(t = 0)$, we will see two unbinding fermions at the left boundary as shown in Fig. 5, which implies the system possesses zero-energy edge states. However, these zero-energy edge states are trivial because they can be washed off after turning on $t$, as shown in Fig. 6. In fact, the existence of trivial zero-energy edge states is nothing too surprising. A chiral-respecting adiabatic deformation can turn a pair of non-zero-energy edge states, where one is $|\psi_1\rangle$ with energy $E$ and the other is $(\mathbb{1}_{N_c \times N_c} \otimes S) |\psi_1\rangle$ with energy $-E$, into two zero-energy edge states without undergoing a phase transition [14, 20].

To illustrate why boundary conditions don't always generate the largest number of linearly independent equations, we apply our approach to this example. For the right semi-infinite limit, the left zero-energy edge states are determined by eq. (60), such as

$$
\begin{pmatrix} A & B \\ \mathbb{1} & 0 \end{pmatrix} L_{Y,j} = \begin{pmatrix} -C & 0 \\ 0 & \mathbb{1} \end{pmatrix} L_{Y,j+1}, \qquad \text{with} \quad |Y, -1\rangle = 0,
$$
$$
\begin{pmatrix} A^\dagger & C^\dagger \\ \mathbb{1} & 0 \end{pmatrix} L_{X,j} = \begin{pmatrix} -B^\dagger & 0 \\ 0 & \mathbb{1} \end{pmatrix} L_{X,j+1}, \quad \text{with} \quad |X, -1\rangle = 0,
$$

(B.3)

where

$$
A = \begin{pmatrix} 0 & t \\ 1 & 0 \end{pmatrix}, \quad B = \begin{pmatrix} 1 & 0 \\ 0 & 0 \end{pmatrix}, \quad C = \begin{pmatrix} 0 & 0 \\ 0 & 1 \end{pmatrix}.
$$

(B.4)

The solutions of the above equations without boundary conditions are given by

$$
L_{Y,j} = c_1 \left[ L_{Y,0} = \begin{pmatrix} 0 \\ 0 \\ 0 \\ 1 \end{pmatrix} \right] + c_2 \left[ L_{Y,0} = \begin{pmatrix} 0 \\ 1 \\ -t \\ 0 \end{pmatrix} + L_{Y,1} = \begin{pmatrix} 0 \\ 0 \\ 0 \\ 1 \end{pmatrix} \right],
$$
$$
L_{X,j} = c_3 \left[ L_{X,0} = \begin{pmatrix} 0 \\ 0 \\ 1 \\ 0 \end{pmatrix} \right] + c_4 \left[ L_{X,0} = \begin{pmatrix} 1 \\ 0 \\ 0 \\ -t \end{pmatrix} + L_{X,1} = \begin{pmatrix} 0 \\ 0 \\ 1 \\ 0 \end{pmatrix} \right].
$$

(B.5)

After considering boundary conditions, it's clear that $c_1 = c_2 = c_3 = c_4 = 0$ for $t \neq 0$. However, if $t = 0$, we have the non-trivial solutions

$$
L_{Y,j} = c_2 \left[ L_{Y,0} = \begin{pmatrix} 0 \\ 1 \\ 0 \\ 0 \end{pmatrix} + L_{Y,1} = \begin{pmatrix} 0 \\ 0 \\ 0 \\ 1 \end{pmatrix} \right] = c_2 \left[ |Y, 0\rangle = \begin{pmatrix} 0 \\ 1 \end{pmatrix} \right],
$$
$$
L_{X,j} = c_4 \left[ L_{X,0} = \begin{pmatrix} 1 \\ 0 \\ 0 \\ 0 \end{pmatrix} + L_{X,1} = \begin{pmatrix} 0 \\ 0 \\ 1 \\ 0 \end{pmatrix} \right] = c_4 \left[ |X, 0\rangle = \begin{pmatrix} 1 \\ 0 \end{pmatrix} \right].
$$

(B.6)

The above discussion indicates that when $t \neq 0$, there is no zero-energy edge state, and when $t = 0$, there are two left zero-energy edge states, where one is located on $|S_X, 0\rangle$ and the other

one is situated on $|S_Y, 0\rangle$, which agrees with the numerical results. Note that we can transform $L_{X/Y,j}$ into $|X/Y, j\rangle$ by the definition $L_{X/Y,j} = \{|X/Y, j\rangle, |X/Y, j-1\rangle\}^T$.

For the left semi-infinite limit, the right zero-energy edge states are given by solving

$$
\begin{pmatrix} A & C \\ \mathbb{1} & 0 \end{pmatrix} L_{Y,j} = \begin{pmatrix} -B & 0 \\ 0 & \mathbb{1} \end{pmatrix} L_{Y,j+1}, \qquad \text{with} \quad |Y, -1\rangle = 0,
$$
$$
\begin{pmatrix} A^\dagger & B^\dagger \\ \mathbb{1} & 0 \end{pmatrix} L_{X,j} = \begin{pmatrix} -C^\dagger & 0 \\ 0 & \mathbb{1} \end{pmatrix} L_{X,j+1}, \qquad \text{with} \quad |X, -1\rangle = 0,
$$
(B.7)

The solutions of the above equations without boundary conditions are

$$
L_{Y,j} = c_1 \left[ L_{Y,0} = \begin{pmatrix} 0 \\ 0 \\ 1 \\ 0 \end{pmatrix} \right] + c_2 \left[ L_{Y,0} = \begin{pmatrix} 1 \\ 0 \\ 0 \\ -1 \end{pmatrix} + L_{Y,1} = \begin{pmatrix} 0 \\ 0 \\ 1 \\ 0 \end{pmatrix} \right],
$$
$$
L_{X,j} = c_3 \left[ L_{X,0} = \begin{pmatrix} 0 \\ 0 \\ 0 \\ 1 \end{pmatrix} \right] + c_4 \left[ L_{X,0} = \begin{pmatrix} 0 \\ 1 \\ -1 \\ 0 \end{pmatrix} + L_{X,1} = \begin{pmatrix} 0 \\ 0 \\ 0 \\ 1 \end{pmatrix} \right].
$$
(B.8)

We can see the above solutions are $t$-independent and after introducing the boundary conditions, there is no non-trivial solution. In other words, no matter how much $t$ is, there is no right zero-energy edge state. It also aligns with the numerical results.

## C  Eigenvalues of the matrix pencil $M_Y - \zeta D_Y$

Let's start with the following block matrix

$$
\begin{pmatrix} A_1 & A_2 \\ A_3 & A_4 \end{pmatrix}.
$$
(C.1)

If $A_4$ is invertible, we have

$$
\det\left[ \begin{pmatrix} A_1 & A_2 \\ A_3 & A_4 \end{pmatrix} \right] = \det[A_4]\det[A_1 - A_2 A_4^{-1} A_3].
$$
(C.2)

We can bring $M_Y - \zeta D_Y$ into the above form, such as

$$
M_Y - \zeta D_Y = \left(
\begin{array}{c|ccccccc}
C_{n_C-1} + \zeta C_{n_C} & C_{n_C-2} & \cdots & C_1 & A & B_1 & \cdots & B_{n_B-1} & B_{n_B} \\
\hline
\mathbb{1} & -\zeta\mathbb{1} & \cdots & \cdots & \cdots & \cdots & \cdots & & 0 \\
0 & \mathbb{1} & -\zeta\mathbb{1} & \cdots & \cdots & \cdots & \cdots & & 0 \\
\vdots & & \ddots & \ddots & & & & & \vdots \\
\vdots & & & \ddots & \ddots & & & & \vdots \\
\vdots & & & & \ddots & \ddots & & & \vdots \\
\vdots & & & & & \ddots & \ddots & & \vdots \\
\vdots & & & & & & \ddots & \ddots & \vdots \\
0 & \cdots & \cdots & \cdots & \cdots & \cdots & 0 & \mathbb{1} & -\zeta\mathbb{1}
\end{array}
\right)
$$
$$
= \begin{pmatrix} A_1 & A_2 \\ A_3 & A_4 \end{pmatrix},
$$
(C.3)

with

$$
\begin{aligned}
A_1 &= C_{n_C-1} + \zeta C_{n_C} \in \mathbb{C}^{n \times n}, \\
A_2 &= \{C_{n_C-2}, \dots, B_{n_B}\} \in \mathbb{C}^{n \times [n \cdot (N_{CB}-1)]}, \\
A_3 &= \{\mathbb{1}, 0, \dots, 0\}^T \in \mathbb{C}^{[n \cdot (N_{CB}-1)] \times n}, \\
A_4 &= T_n(\mathbb{1}, -\zeta \mathbb{1}, 0) \in \mathbb{C}^{[n \cdot (N_{CB}-1)] \times [n \cdot (N_{CB}-1)]},
\end{aligned}
\tag{C.4}
$$

where $T_n$ denotes the block tridiagonal Toeplitz matrix and $N_{CB} = n_C + n_B$. Because $A_4$ is a lower triangular matrix, the determinant of $A_4$ is given by the product of the diagonal entries,

$$
\det[A_4] = (-\zeta)^{[n \cdot (N_{CB}-1)]}.
$$

The inverse of $A_4$ is

$$
A_4^{-1} = \begin{pmatrix}
-\zeta^{-1}\mathbb{1} & 0 & 0 & 0 & 0 \\
-\zeta^{-2}\mathbb{1} & -\zeta^{-1}\mathbb{1} & 0 & 0 & 0 \\
-\zeta^{-3}\mathbb{1} & -\zeta^{-2}\mathbb{1} & -\zeta^{-1}\mathbb{1} & 0 & 0 \\
\vdots & \vdots & \vdots & \ddots & 0 \\
-\zeta^{-(N_{CB}-1)}\mathbb{1} & -\zeta^{-(N_{CB}-2)}\mathbb{1} & -\zeta^{-(N_{CB}-3)}\mathbb{1} & \cdots & -\zeta^{-1}\mathbb{1}
\end{pmatrix}.
$$

With all these in mind, we obtain

$$
\begin{aligned}
\det[M_Y - \zeta D_Y] &= \det\left[\begin{pmatrix} A_1 & A_2 \\ A_3 & A_4 \end{pmatrix}\right] \\
&= (-\zeta)^{[n \cdot (N_{CB}-1)]} \det[\zeta C_{n_C} + C_{n_C-1} + \zeta^{-1} C_{n_C-2} + \cdots + \zeta^{2-n_C} C_1 \\
&\qquad\qquad + \zeta^{1-n_C} A + \zeta^{-n_C} B_1 + \cdots + \zeta^{-(N_{CB}-1)} B_{n_B}] \\
&= (-1)^{[n \cdot (N_{CB}-1)]} \det[\zeta^{N_{CB}} C_{n_C} + \zeta^{(N_{CB}-1)} C_{n_C-1} + \zeta^{(N_{CB}-2)} C_{n_C-2} \\
&\qquad\qquad + \cdots + \zeta^{n_B+1} C_1 + \zeta^{n_B} A + \zeta^{n_B-1} B_1 + \cdots + B_{n_B}] \\
&= (-1)^{[n \cdot (N_{CB}-1)]} \det\left[A\zeta^{n_B} + \sum_{m'=1}^{n_B} B_{m'}\zeta^{n_B-m'} + \sum_{n'=1}^{n_C} C_{n'}\zeta^{n'+n_B}\right] \\
&= (-1)^{[n \cdot (n_C+n_B-1)]} g_2(\zeta).
\end{aligned}
\tag{C.5}
$$

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
