# Peer review of "A linear algebra-based approach to understanding the relation between the winding number and zero-energy edge states"

_SciPost Physics Core, doi:SciPost Phys. Core 7, 003 (2024)_

## Round 3 · Referee Report · Anonymous (Referee 1) · 2024-1-10

Report
The author considers gapped chiral-symmetric 1D systems with an arbitrary number of bands and arbitrary but finite range hopping. They prove that the winding number is equal to the number of robust edge modes at either end of the chain, thus establishing bulk-boundary correspondence. This is done while avoiding advanced mathematical techniques, and resorting instead to basic algebra and complex analysis.
I found the work to be valid and well-presented. The paper is well-structured, starting with simple cases and gradually moving towards more complex systems. I believe that this work should be published with only minor changes, but I think it makes a better fit to Scipost Physics Core rather than Scipost Physics. The reason for this is that I consider it to be mainly a follow-up work, using a methodology similar to that of Ref. [28] and extending those previous results to the case of systems with more than two bands.
The only two minor comments I have are:
1) just above Eq. 1, I believe "antiunitary" should be replaced with "unitary"
2) In Section 6, I believe it might help the reader if there was an intuitive explanation for the main result of that section. Maybe something along the lines of: since the zero mode lives only on the B sublattice, it cannot be altered by any change occurring only on the A sublattice sites, even if that change breaks chiral symmetry.
Author: Chen-Shen Lee on 2024-01-31 [id 4297]
(in reply to Report 1 on 2024-01-10)We thank the referee for the careful review and the valuable suggestions. In the final version, we have addressed the points raised in the report. 1. Thank you sincerely for pointing out this error. The chiral operator is unitary in the single-particle basis. 2. we added an intuitive explanation followed by eq. (90).
Besides these, we found a non-obvious ambiguity concerning the notation |X/Y, j〉. We clarified it by introducing -) $|S_X/S_Y, j〉= |S_X/S_Y〉\otimes |j〉$ where $|S_X/S_Y〉$ is the basis state of the chiral operator with eigenvalue +1/-1. -) |X/Y, j〉: the vector composed of the coefficients according to the basis $|S_X/S_Y, j〉$

---

## Editorial Decision

published